# Enhanced Quinolone Resistance and Differential Expression of Efflux Pump *nor* Genes in *Staphylococcus aureus* Grown in Platelet Concentrates

**DOI:** 10.3390/antibiotics14070635

**Published:** 2025-06-21

**Authors:** Carina Paredes, Que Chi Truong-Bolduc, Yin Wang, David C. Hooper, Sandra Ramirez-Arcos

**Affiliations:** 1Department of Biochemistry, Microbiology, and Immunology (BMI), Faculty of Medicine, University of Ottawa, Ottawa, ON K1G 4J5, Canada; carina.paredes@blood.ca; 2Donation Policy & Studies, Canadian Blood Services, Ottawa, ON K1G 4J5, Canada; 3Division of Infectious Diseases, Massachusetts General Hospital, Harvard Medical School, Boston, MA 02114, USA; qtruongbolduc@mgh.harvard.edu (Q.C.T.-B.); ywang53@mgb.org (Y.W.); dhooper@mgh.harvard.edu (D.C.H.)

**Keywords:** *Staphylococcus aureus*, platelet concentrates, quinolone resistance by *Staphylococcus aureus*, *Staphylococcus aureus* virulence, silkworm animal model

## Abstract

**Background/Objective**: Platelet concentrates (PCs) are used in transfusion medicine to treat bleeding disorders. *Staphylococcus aureus*, a predominant PC contaminant, has been implicated in several adverse transfusion reactions. The aim of this study was to investigate the impact of PC storage on *S. aureus* resistance to quinolones, which are commonly used to treat *S. aureus* infections. **Methods/Results**: Four transfusion-relevant *S. aureus* strains (TRSs) were subjected to comparative transcriptome analyses when grown in PCs vs. trypticase soy broth (TSB). Results of these analyses revealed differentially expressed genes involved in antibiotic resistance. Of interest, the *norB* gene (encodes for the NorB efflux pump, which is implicated in quinolone resistance and is negatively regulated by MgrA) was upregulated (1.2–4.7-fold increase) in all PC-grown TRS compared to TSB cultures. Minimal Bactericidal Concentration (MBC) of ciprofloxacin and norfloxacin in PC-grown TRS compared to TSB showed increased resistance to both quinolones in PC cultures. Complementary studies with non-transfusion-relevant strains *S. aureus* RN6390 and its *norB* and *mgrA* deletion mutants were conducted. MBC of ciprofloxacin and norfloxacin and RT-qPCR assays of these strains showed that not only *norB*, but also *norA* and *norC* may be involved in enhanced quinolone resistance in PC-grown *S. aureus*. The role of *norB* in *S. aureus* virulence was also tested using the silkworm *Bombyx mori* animal model; lethal dose 50 (LD_50_) assays revealed slightly higher virulence in larvae infected with the wild-type strain compared to the *norB* mutant. **Conclusions**: The PC storage environment enhances quinolone resistance in *S. aureus* and induces differential expression of efflux pump *nor* genes. Furthermore, our preliminary data of the involvement of NorB in virulence of *S. aureus* using a silkworm model merit further investigation with other systems such as a mammal animal model. Our results provide mechanistic insights to aid clinicians in the selection of antimicrobial treatment of patients receiving transfusions of *S. aureus*-contaminated PCs.

## 1. Introduction

Platelet concentrates (PCs) are used to treat patients with bleeding disorders and consist of platelets suspended in plasma or a mix of plasma and platelet additive solution [1]. Platelet function and quality is maintained in the well-established PC storage conditions including incubation at 20–24 °C under constant agitation. Although important to maintaining platelet functionality, these conditions also sustain the growth of bacteria that may enter the blood collection bag during venipuncture [2]. To minimize the risk of transfusing bacterially contaminated PCs, Canadian Blood Services employ several strategies, including donor skin disinfection, diversion of the first 30–40 mL of collected blood, PC screening using the BACT/ALERT 3D automated culture system, and, more recently, PC treatment with the pathogen reduction technology INTERCEPT [2,3].

PC screening with the BACT/ALERT system is a gold standard method to improve PC safety; unfortunately, bacteria such as *Staphylococcus aureus* and *Staphylococcus epidermidis* can escape detection, posing a risk to transfusion recipients [3,4,5]. Recent research studies have demonstrated that both staphylococcal species grown in PCs exhibit increased expression of virulence-associated genes [6,7]. Additionally, *S. epidermidis* displays enhanced virulence when grown in PCs compared to laboratory media using the *Caenorhabditis elegans* animal model [8].

*S. aureus* is a facultative aerobic Gram-positive bacterium naturally present in the skin and mucosa of healthy humans and responsible for a wide range of community- and healthcare-associated infections [9]. Importantly, *S. aureus* is a predominant PC contaminant which has been involved in septic transfusion reactions worldwide and has therefore become an important safety risk to PC transfusion recipients due to its ability to escape detection during routine PC screening, grow to clinically significant levels, and produce exotoxins during PC storage [5,9].

Furthermore, *S. aureus* is known as a superbug due to the difficulty of treating infections caused by this organism with antimicrobial drugs [9]. There are several mechanisms involved in antimicrobial resistance in *S. aureus* including modifications of drug targets, drug inactivation, or drug extrusion via efflux pumps [10,11,12]. In Canada, quinolones are used to treat *S. aureus* infections [13]. However, this species has developed mechanisms of quinolone resistance. One of the strategies that *S. aureus* uses to resist quinolones involves target alteration. Specifically, mutations in the *gyrA* and *gyrB* genes which encode for the DNA gyrase or in the *parC* and *parE* genes which encode for the topoisomerase IV result in decreased binding affinity to quinolones [12]. Furthermore, the Major Facilitator Superfamily (MFS) NorA, NorB, and NorC efflux pumps are involved in resistance to quinolones such as norfloxacin and ciprofloxacin [14]. As illustrated in Figure 1, MgrA is phosphorylated by the PknB kinase [15,16]; while *norA* expression is repressed by MgrA in its dephosphorylated state, *norB* is repressed when phosphorylated MgrA (MgrA-P) binds to its promoter [15,16,17,18]. Additionally, MgrA can be dephosphorylated by the RsbU phosphatase [16]. RsbU has a major role in the alternative sigma factor SigB regulon, which consists of four proteins, RsbV, RsbW, RsbU, and SigB [19]. When the bacterium is exposed to a stressful environment, RsbU dephosphorylates RsbV-P; dephosphorylated RsbV then binds to RsbW releasing SigB to form a complex with RNA polymerase resulting in the holoenzyme RNAP [19,20] (Figure 1). This holoenzyme drives the transcription of genes involved in housekeeping functions, virulence, biofilm formation, persistence, cell internalization, membrane transport, and antimicrobial resistance [20].

In addition to antibiotic resistance, *S. aureus* produces several virulence factors including exotoxins, immune evasion factors, and membrane proteins that facilitate bacterial infection, making this bacterium a major threat for PC recipients [9]. Several animal models, including *C. elegans*, have been used to study *S. aureus* virulence in settings different from the PC storage environment [21]. Invertebrate animal models are amenable to investigate bacterial virulence and antimicrobial resistance as they are genetically simpler compared to other animal models, have low cost, and allow for a large sample size in research studies [22,23]. Within the invertebrate animal models, the *Bombyx mori* silkworm has been used to test *S. aureus* resistance to antimicrobial peptides [24]. More recently protocols have been established to test the virulence of the transfusion relevant bacterium *Cutibacterium acnes* using *B. mori* larvae [25].

Previous studies have demonstrated that *Staphylococcus* species grown in PCs display upregulation of genes encoding for antimicrobial resistance and virulence. A study from Loza et al. [7] has shown that exposure to the storage conditions and selective pressures present in PCs can lead to upregulation of genes associated with antibiotic resistance and enhanced virulence traits of *S. epidermidis.* Moreover, it has been shown that the PC environment triggers the upregulation of several virulence genes in PC-grown *S. aureus* compared to media [6,9]. These findings suggest that the PC environment provides a niche that not only supports staphylococcal survival but may also drive the expression of genes that contribute to pathogenicity and resistance to antibiotic treatment. The molecular mechanisms by which PC induces changes in bacterial virulence factors, including antibiotic resistance genes, are unknown.

Our previous studies have provided evidence of differential expression of antibiotic resistance genes in *S. aureus* grown in PCs [6,9]. To expand the current knowledge, this study aimed to directly test whether the increased gene expression observed under PC conditions translates into heightened antibiotic resistance with focus on resistance to quinolones by conducting comparative minimal bactericidal concentration assays. Furthermore, the work presented herein explored the role of the NorB efflux pump on virulence of *S. aureus* using a silkworm animal model.

## 2. Results

### 2.1. The PC Storage Milieu Triggers Upregulation of the norA, norB, and norC Efflux Pumps in a Strain-Dependent Manner

Differential expression of virulence factors between staphylococci grown in PCs and media have shown that genes involved in antibiotic resistance are upregulated in PC-grown *S. epidermidis* [7], and therefore we tested whether a similar phenotype was observed in *S. aureus*. Four transfusion-relevant *S. aureus* strains (TRSs), CBS2016-05, CI/BAC/25/13/W, PS/BAC/169/17/W, and PS/BAC/317/16/W) [26,27,28,29] were used for this study (Table 1). *S. aureus* CBS2016-05 was involved in a false-negative screening septic transfusion event in Canada while *S. aureus* CI/BAC/25/13/W was isolated in the National Health Service Blood and Transplant (NHSBT) in England, from a unit with false-negative screening results that presented aggregates and was therefore not transfused (near-miss). The other two strains, *S. aureus* PS/BAC/169/17/W and *S. aureus* PS/BAC/317/16/W, were detected during routine PC screening at the NHSBT. All isolates were grown in PCs and trypticase soy broth (TSB) and then subjected to comparative transcriptome analyses which showed differential regulation of antimicrobial resistance genes. This analysis revealed differentially expressed genes involved in various mechanisms of resistance to antibiotics, including the Major Facilitator Superfamily (MFS), the Multidrug and toxin extrusion family (MATE), ATP-binding cassette superfamily (ABC), and genes encoding for enzymes that modify the cellular targets of antibiotics and enzymes that modify or metabolize antimicrobial drugs (Figure 2).

The *norB* gene was the only one that was upregulated in the four TRSs with a significant log2 fold change ranging from 1.23 to 4.71 when *S. aureus* was grown in PCs compared to TSB (Figure 2). However, the other efflux pump genes of the *nor* family, *norA* and *norC*, were either downregulated or not differentially expressed (i.e., log2 fold change = 0) in PC-grown TRS compared to TSB cultures (Figure 2). The *norB* gene encodes for the efflux pump NorB, which is involved in quinolone resistance and is negatively regulated by MgrA [16,17,18]. Transcriptome data were verified with RT-qPCR assays, which confirmed the upregulation of *norB* in PC-grown *S. aureus* compared to TSB cultures in all four TRSs (Figure 3). Therefore, the role of NorB in *S. aureus* resistance to quinolones was further investigated using non-transfusion-relevant strains which included wild-type *S. aureus* RN6390 and its derivative mutants *norB* and *mgrA*. Interestingly, RT-qPCR showed that expression of all three genes, *norA*, *norB*, and *norC*, was increased (one to six-fold) in *S. aureus* RN6390 when this strain was grown in PCs and expression of both *norA* and *norC* genes was enhanced in PC cultures of the *norB* mutant (Figure 4). Unexpectedly, only *norB* was upregulated in the *mgrA* mutant despite that the global regulator MgrA represses expression of all three *nor* genes. Downregulation of *norA* and *norC* in the *mgrA* mutant strain could be due to the action of other regulators such as NorG, which is known to negatively regulate NorC [32] and the two-component regulatory system ArlR-ArlS, which is a negative regulator of NorA [33]. Importantly, in wild-type *S. aureus* RN6390, MgrA represses expression of NorG; therefore, in its absence, NorG can be active. Furthermore, NorG is a positive regulator of NorB and ArlS [17]. This complex interaction between regulators supports the findings presented in Figure 4 for the *mgrA* mutant. However, further experiments are needed to confirm this hypothesis and test the expression of other regulators in a *mgrA*- background. The differences observed in *nor* gene expression between TRS and RN6390 strains indicate that gene regulation of *S. aureus* grown in PCs depends on the strain’s genetic background.

### 2.2. S. aureus Grown in PCs Displays Heightened Resistance to Quinolones

In Canada, quinolones are included as part of the treatment regimen for infections caused by *S. aureus.* [13]. Therefore, ciprofloxacin and norfloxacin were chosen to test quinolone resistance in PC-grown *S. aureus*. Minimal Inhibitory Concentration (MIC) and Minimal Bactericidal Concentration (MBC) assays for both antibiotics were determined for the four TRSs, *S. aureus* RN6390, its derivative *norB* and *mgrA* mutants, and control *S. aureus* ATCC 29213. MIC data are shown in Appendix A. For MBC assays, all strains were cultured in PCs and TSB. MBC values in TSB were comparable to MIC results (Table 2 and Appendix A); however, MBC results for both quinolones were higher in PC-grown bacteria compared to TSB cultures reaching significant difference for ciprofloxacin (*p* < 0.05) in all strains tested except *S. aureus* PS/BAC/317/16/W, which was marginally different (*p* = 0.057) (Table 2). While only increased expression of *norB* may be associated with enhanced quinolone resistance in the TRS grown in PCs (Table 2 and Figure 2), all three *norA*, *norB,* and *norC* genes were upregulated in *S. aureus* RN6390 (Figure 4), which could play a role in increased quinolone resistance in this strain (Table 2). Comparative genomic analyses between TRSs and *S. aureus* RN6390 revealed no major differences in the genomic content and structure (Appendix A, Appendix A) except for the mutation in the phosphatase *rsbU* gene in *S. aureus* RN6390. Therefore, the differences in *nor* gene expression between TRSs and the RN6390 strain, and consequent quinolone resistance, is likely due to the *rsbU-* background of *S. aureus* RN6390. It is anticipated that a mutation in the *rsbU* gene, which encodes for the RsbU phosphatase involved in MgrA dephosphorylation [16], would alter the ratio between MgrA and MgrA-P (Figure 1), resulting in differential expression of the *nor* genes, which warrants further studies.

Importantly, MBC values for ciprofloxacin and norfloxacin in wild-type *S. aureus* RN6390 and the *norB* mutant grown in PCs were not significantly different (*p* = 0.065 and *p* = 0.27, respectively, Table 2) indicating that *norA* and *norC*, which are upregulated in the *norB* mutant (Figure 4), confer resistance to these quinolones in a *norB-* background. However, resistance to ciprofloxacin and norfloxacin was significantly higher in the *mgrA* mutant compared to the wild-type strain (*p* = 0.0002 and *p* = 0.0011, respectively; Table 2), due mostly to the overexpression of *norB* in this strain as shown in Figure 4.

### 2.3. NorB Is Involved in Virulence of S. aureus in a Silkworm Model

It has been shown that the PC storage environment enhances the virulence of transfusion-relevant *S. aureus* CBS2016-05 and that overexpression of NorB increases *S. aureus* survival in a mouse abscess model [25,34]. With this information, we hypothesized that NorB could be involved in increased growth in PCs and enhanced virulence of *S. aureus*.

As shown in Figure 5, the growth dynamics of wild-type *S. aureus* RN6390 was not different from the growth displayed by the *norB* and *mgrA* mutants in PCs; therefore, *norB* does not seem to confer a growth advantage to *S. aureus* during PC storage.

In virulence studies using animal models, lethal dose (LD_50_) values indicate the bacterial load needed to kill 50% of the population [35]. In this study, three groups of 10 *B. mori* larvae per bacterial isolate (*S. aureus* RN6390 and *norB* and *mgrA* mutants) were injected with 10-fold bacterial suspensions to determine the bacterial load that kills 50% of each group. LD_50_ studies in *B. mori* larvae showed that the *S. aureus norB* mutant (RN6390∆*norB*) had a higher LD_50_ than the wild-type strain. Two-log more of *S. aureus* RN6390∆*norB* were needed to kill 50% of larvae compared to wild-type *S. aureus* RN6390, indicating a decrease in the virulence of the mutant strain (Table 3). Although no statistical significance was attained (*p* > 0.05), biologically, a difference in 2-log of bacteria is meaningful indicating that NorB could be involved in resistance to immune clearance probably by exporting silkworm immune factors such as antimicrobial peptides from staphylococcal cells. Efflux pumps are involved in resistance to antimicrobial peptides in other bacteria such as *Pseudomonas aeruginosa* [36]. This provides novel evidence of the NorB role in *S. aureus* virulence. As *norB* is overexpressed in the *S. aureus mgrA* mutant (Figure 4), it was expected to observe increased virulence of this mutant compared to the wild-type strain. However, LD_50_ values of the *mgrA* mutant were higher than those of the wild-type strain. As MgrA is a global regulator, the absence of the *mgrA* gene can trigger the repression of virulence genes that would be normally expressed in the wild-type strain.

## 3. Discussion

This study validated previous gene expression studies that indicated that the PC storage environment triggers increased antibiotic resistance in contaminant bacteria. We demonstrated that quinolone resistance significantly increased in *S. aureus* grown in PCs compared to media, independently of the genetic background of the strains. Furthermore, novel evidence of the role of the efflux pump NorB in *S. aureus* virulence was provided using a silkworm model organism.

PC storage at room temperature under agitation safeguards platelet functionality, which is critical for therapeutic efficacy. Unfortunately, these conditions are amenable for growth of most bacterial contaminants introduced during venipuncture [3]. Storage of PCs is limited to a maximum of seven days to minimize the platelet storage lesion (PSL), which encompasses a series of morphological and metabolic changes that platelets undergo as a result of an increase in lactate levels, with consequential decrease in pH, loss of surface membrane glycoproteins, and reduced aggregation response [37,38]. A study by Yousuf et al. showed that *S. aureus* proliferation to clinically significant levels in PCs enhances the PSL [6]. Importantly, changes in the PC storage environment not only affect platelet metabolism but also impact gene expression of bacterial contaminants. Immune stressors released by activated platelets during PC storage trigger differential expression of virulence genes such as those involved in biofilm formation and antimicrobial resistance in *S. aureus* and *S. epidermidis* [6,7,39].

*S. aureus* is the predominant PC contaminant involved in septic transfusion reactions worldwide [9]. In this study, we demonstrated that different isolates of *S. aureus*, including transfusion-relevant strains, displayed increased MBC values to the fluoroquinolones ciprofloxacin and norfloxacin when the bacterium was grown in PCs compared to TSB cultures. Transcriptome and RT-qPCR analyses of four transfusion-relevant strains showed upregulation of the *norB* efflux pump gene, which was associated with resistance to quinolones in all tested strains grown in PCs versus TSB. Importantly, genes encoding for other efflux pumps such as *norA* and *norC* were not differentially regulated in transfusion-relevant strains. However, upregulation of *norA*, *norB,* and *norC* was observed in *S. aureus* RN6390 strains suggesting that NorA and NorC may also participate in increased quinolone resistance in PC cultures in a strain-dependent manner.

Previous studies have shown that overexpression of *S. aureus norA* is directly related to increased resistance to ciprofloxacin [40,41,42]. Nevertheless, our results should be analyzed with caution as *S. aureus* RN6390 lacks the RsbU phosphatase [30], which is involved in dephosphorylation of the global regulator MgrA [16]. It is intriguing to investigate whether other phosphatases (bacterial or platelet-derived) can change the phosphorylation status of MgrA and whether consequent expression of *nor* genes during PC storage, as repression of *norA* and *norB* depends on the phosphorylation state of MgrA [20]. The quinolone resistance results presented herein are likely the consequence of complex interactive regulatory processes that merit further investigation.

It has been shown that NorB favors the growth of *S. aureus* at low pH [43], which is one of the characteristics of the PSL as mentioned above. However, no differences in bacterial proliferation were observed between wild-type *S. aureus* RN6390 and its *norB* mutant during PC storage, indicating that NorB does not confer advantageous *S. aureus* proliferation in PCs.

In addition to antibiotic resistance, overexpression of NorB confers advantageous survival of *S. aureus* in skin abscesses [34]. The PC storage environment also poses challenges for *S. aureus* survival due to the presence of immune stressors [9]. Furthermore, the virulence of the transfusion-relevant strain *S. aureus* CBS2016-05 is enhanced when grown in PCs compared to media as recently demonstrated using a silkworm animal model [25]. We therefore tested the role of *norB* in virulence of non-transfusion-relevant strains of *S. aureus* using silkworms. Our data showed a 2-log higher LD_50_ for the *S. aureus norB* mutant compared to wild-type *S. aureus* RN6390, indicating loss of virulence. It is important to recognize that the *rsbU*-negative background of the RN6390 strains may have contributed to our observations in silkworms, which warrants more extensive studies using different *S. aureus* strains, animal models, or cell cultures [44,45,46]. As RsbU has a major role in the SigB regulon, it is also critical to consider how the expression of other virulence factors driven by SigB played a role in the virulence results presented herein. This can be studied by performing comparative assays between *S. aureus* RN6390 and a *rsbU+* strain such as *S. aureus* SH1000. We propose that NorB may be involved in excreting antimicrobial peptides produced by silkworm larvae during infection as shown for other efflux pumps in *P. aeruginosa* [36]. It would therefore be interesting to investigate how expression of *norB* in a *rsbU+* background influences virulence in silkworms as overexpression of *sigB* has been associated with in vitro resistance to antimicrobial peptides produced by nematodes in *S. aureus* [47].

We provided novel information on the complex regulation of *S. aureus nor* efflux pump genes in contaminated PCs. Additionally, we showed evidence of the potential role of *norB* in *S. aureus* virulence using a silkworm model. Our findings highlight the need to deepen our knowledge on the molecular mechanisms involved in resistance to antibiotics and virulence when bacteria proliferate in PCs. Understanding the complex processes involved in platelet–bacteria interactions in the unique PC storage environment could be used to propose mechanisms for the prevention of septic transfusion reactions involving *S. aureus*-contaminated PCs.

### Recommendations for Further Investigation

Downregulation of *norA* and *norC* in the *mgrA* mutant strain could be due to the action of other regulators which could be investigated by performing gene expression studies in *mgrA* mutant strains compared to their parental wild-type counterparts.We discussed that our MBC data depend on the expression of efflux pump *nor* genes in different bacterial backgrounds and potentially the phosphorylation status of the global regulator MgrA. Investigating changes in phosphorylation of MgrA when *S. aureus* is grown in PCs and consequent expression of *nor* genes will provide new insight to advance the interpretation of our data.Our virulence experiments were conducted in *S. aureus* RN6390, which has a negative *sigB* phenotype due to a mutation in *rsbU.* Therefore, it would be interesting to create a *norB* mutant in a strain with *rsbU+* background and study the role of *norB* in virulence with a functional SigB regulon.We showed that the silkworm model was an appropriate tool to obtain preliminary data on the role of *norB* in virulence. However, these results could be complemented with assays that reflect human immune response such as experiments using human cell lines or mammalian animal models.Our hypothesis of the role on NorB in exporting antimicrobial peptides from cells of infected silkworm larvae could be tested by performing in vitro experiments to test resistance to antimicrobial peptides produced by silkworms using *S. aureus* with different genetic backgrounds.

## 4. Materials and Methods

### 4.1. Bacterial Strains, Plasmids, and Growth Conditions

The origin of *S. aureus* strains CBS2016-05, CI/BAC/13/W, PS/BAC/169/17/W, PS/BAC/317/16/W, RN6390, RN6390Δ*norB*, RN6390Δ*mgrA,* and ATCC 29213 is described in Table 1. *S. aureus* isolates were routinely cultured on trypticase soy agar (TSA) for colony isolation or trypticase soy broth (TSB) and incubated with agitation at 20–24 °C for 6 days, or static at 37 °C for 24 h. In PCs, the strains were grown with agitation at 20–24 °C for 6 days (PC storage conditions). All bacterial strains were stored in a brain–heart infusion broth with 15% glycerol (*v*/*v*) at −80 °C. TSA was used to sub-culture frozen stocks and for bacterial enumeration.

### 4.2. Pooled Platelet Concentrates

Leukocyte-reduced buffy coat pooled platelet concentrates suspended in 100% plasma were used for this study following standard manufacturing protocols established at the Canadian Blood Services netCAD Blood4Research Facility (netCAD, Vancouver, BC, Canada). This study was granted ethical approval by the Canadian Blood Services Research Ethical Board (REB 2015.024, 24 June 2019).

### 4.3. Comparative Genomic Analyses

The genomes of five *S. aureus* isolates, the four TRS and RN6390, which were previously published [26,27,28,29,30], were aligned and visualized using Mauve Progressive (Mauve 2015226 built10, assessed on 8 April 2025). Genome files (FASTA format) were uploaded onto the software for comparison using a guide tree constructed from pairwise comparisons to identify conserved segments, insertions, deletions, and rearrangements within the genomes. Genome inversion features were visualized using Proksee (assessed on 10 April 2025).

### 4.4. Transcriptome Analyses (PCs vs. TSB) and Selection of Differentially Expressed Genes (DEGs)

Three independent *S. aureus* TSB and PC cultures were prepared with an initial inoculum of approximately 4 × 10^6^ CFU/mL. *S. aureus* RNA was isolated from bacterial samples grown to mid-stationary phase following previously established protocols [6,39]. Briefly, total RNA was isolated using the commercial kit FastRNA™ Pro Blue Kit (MP Biologicals, Santa Ana, CA), followed by DNase treatment for purification. RNA extracted from *S. aureus* grown in PCs was subject to a mammalian depletion treatment. RNA samples were sent to the StemCore Laboratories located at the Core Facilities at University of Ottawa and the Ottawa Hospital Research Institute (https://www.ohri.ca/bioinformatics/, accessed on 17 June 2025) for RNA sequencing and differential gene expression (DGE) analysis as described previously [42,43]. RNA samples with RIN > 8 were used to prepare paired-end libraries and sequenced. The libraries were normalized, pooled, and diluted as required to achieve acceptable cluster density on the NextSeq 500 sequencer (Illumina SY-414-1001) (Illumina Inc., Baltimore, MD, US). The transcriptome dataset of antibiotic resistance genes was further analyzed considering a log2-fold difference of ≥1.0 as a significant change in DGE. This cut-off was chosen to capture relatively small but biologically meaningful gene expression changes.

### 4.5. Quantitative Reverse Transcription PCR (RT-qPCR)

RT-qPCR assays were developed following established procedures [6,39]. Total RNA served as a template in the synthesis of cDNA using the QuantiTect RT kit (Qiagen, Germantown, MD, US). Primers (Appendix A) were designed targeting *nor* genes and qPCR was performed using QuantiNova SYBR green PCR kit (Qiagen, Hilden, German) according to manufacturer recommendations, the 16S RNA gene served as the housekeeping gene, and a blank control reaction was conducted for all the samples using ddH_2_O instead of template. Primer efficiency is shown in Appendix A.

### 4.6. Construction of a S. aureus RN6390 norB Deletion Mutant

A *norB* in-frame deletion mutant was prepared following an allelic exchange protocol previously optimized [47]. Briefly, upstream and downstream sequences of the gene/locus to be deleted were amplified by overlay polymerase chain reaction (PCR), and the two fragments were linked by overlay PCR (primers used for this procedure are listed in Appendix A). PCR products and plasmid pIMAY were double-digested with KpnI and SacI, ligated together, transformed into *E. coli* DC10B, and incubated on Luria–Bertani (LB) containing chloramphenicol at 10 μg/mL at 37 °C. The construct pIMAY-Δ*norB* was extracted and transformed into *S. aureus* strain RN4220 and then into *S. aureus* RN6390 for subsequent allelic exchange. Positive RN6390 transformants were cultured at 28 °C on TSA supplemented with chloramphenicol at 10 μg/mL. Each step was verified with DNA sequencing to confirm the plasmid construction. RN6390-positive transformants were diluted between 10- and 1000-fold and plated with chloramphenicol at 10 μg/mL and grown at 37 °C to integrate the construct pIMAY-Δ*norB* into the chromosome. Absence of extrachromosomal plasmid was confirmed by PCR using primers designed from the plasmid pIMAY. Colonies were selected, plated on brain–heart infusion agar plates supplemented with anhydrotetracycline at 1 μg/mL, then grown at 28 °C for 48 h. Chloramphenicol-sensitive colonies were selected and verified by DNA sequencing. The deletion mutants were named RN6390Δ*norB* and the absence of *norB* was verified by DNA sequencing.

### 4.7. Antibiotic Resistance Assays

These assays were performed for all isolates to assess bacterial potential resistance to the fluoroquinolones ciprofloxacin and norfloxacin. *S. aureus* ATCC 29213 was used as a control for MIC and MBC assays. MIC values were determined according to the microdilution broth method described in the Clinical Laboratory Standards Institute (CLSI) guidelines [48]. Antibiotic stock solutions were prepared as per manufacturers’ specifications. Bacterial colonies were re-suspended in cation-adjusted Müeller–Hinton broth (MHB^ca+^) to 0.5 Densimat, which corresponds to approximately 1 × 10^8^ CFU/mL, and then diluted to a bacterial load of approximately 2 × 10^6^ CFU/mL. Bacterial suspensions were distributed into a 96-well plate within 15 min of preparation. The 96-well plates were loaded as follows: wells of the outer rows and columns were filled with 200 μL of MHB^ca+^ and the internal wells were filled with 100 μL of serially diluted antibiotics in MHB^ca+^ to obtain antibiotic concentration gradients across the plates. Antibiotics were added in double concentration since they were diluted 2-fold once the bacterial suspensions were added (e.g., stock solution of 512 μg/mL to ensure a final concentration of 256 μg/mL in the well). Following antibiotic dispensing, 100 μL of the bacterial suspensions were added to each well and were thoroughly mixed. The top only had unspiked MHB^ca+^ (negative control) while the last row was filled with the bacterial suspension without antibiotics (positive control). Plates were incubated for 24 h at 37 °C. After incubation, well absorbance (OD_600_) was recorded using a plate reader and MIC values were determined as the lowest concentration of the antibiotic inhibiting bacterial growth. MBC assays were optimized in collaboration with Dr. Thien-Fah Mah (University of Ottawa) [49]. Briefly, an antibiotic gradient was prepared and 50 μL was pre-loaded onto 48-well plates, overnight bacterial cultures were adjusted to 0.5 Densimat and inoculated in TSB or PCs to approximately 5.5 × 10^5^ CFU/mL, and 450 μL were loaded onto the 48-well plates previously loaded with an antibiotic and incubated. Plates loaded with PCs were incubated at PC conditions (22 ± 2 °C, for 5 days with constant agitation) and plates loaded with inoculated TSB were incubated at optimal conditions for bacterial growth (37 °C, for 24 h without agitation). After incubation, samples from each well were spotted in TSA (~3.5 μL/spot) and incubated overnight at 37 °C. The presence or absence of bacterial growth was recorded to establish the MBC value. MIC and MBC assays were performed three independent times with technical duplicates.

### 4.8. Bacterial Growth Curves

*S. aureus* RN6390 and its derivative *norB* and *mgrA* mutants were cultured in PCs by inoculating units with an approximate initial inoculum of 30 CFU/PC unit to mimic real-life bacterial loads in PCs. Bacterially inoculated PCs were incubated under standard PC storage conditions (22 ± 2 °C, agitation). Samples of 1 mL were taken every 24 h for a total of 5 days for plating in duplicate on TSA plates, which were incubated for 24 h at 37 °C for colony counts and determination of bacterial loads. Growth curves were repeated three independent times.

### 4.9. Silkworm Rearing

Silkworm assays were established following recently optimized protocols [26]. Briefly, *B. mori* eggs were acquired from coastal silk and were kept at room temperature until hatched. Silkworm larvae were fed on commercially bought de-hydrated silkworm chow (coastal silk) supplemented with 300 mg of vancomycin/Kg upon re-hydration. Fifth in-star larvae (day 1 and 2 post molt) were fed food free of vancomycin for a minimum of 24 h prior to use in downstream assays.

### 4.10. Determination of Lethal Dose 50 (LD_50_)

*S. aureus* isolates were grown overnight aerobically at 37 °C with agitation in TSB media. The load required to kill 50% of the silkworm test population (LD_50_) was determined as recently described [26]. Briefly, overnight cultures were centrifuged, resuspended in insect saline (0.6% NaCl), serially diluted (ten-fold), and inoculated into the haemolymph of silkworm larvae (30 μL/10 larvae/dilution). Bacterial load of each suspension was determined by plating on TSA plates followed by incubation at 37 °C for 24 h and colony counting. Silkworm larvae were incubated at 37 °C for three days (72 h) and inspected daily to assess larval survival and death. Insect saline served as control; LD_50_ studies were performed in triplicate.

### 4.11. Statistical Analyses

Statistical analysis of the RNA-seq data was performed in R. Statistical comparisons of MBC values and LD_50_ values were done with GraphPad Prism (version 9.0.0) software using Mann–Whitney U test and T-test with Welch’s correction, respectively. Values were expressed as mean ± SE and a *p*-value of <0.05 was considered statistically significant.

## 5. Conclusions

Our studies aimed to investigate the impact of the PC storage environment on *S. aureus* transcriptional changes with a focus on genes involved in antibiotic resistance. The findings of this study demonstrate that the PC storage environment triggers *S. aureus* resistance to quinolones and upregulation of *nor* efflux pump genes in a strain-dependent manner. Notably, NorB was shown to be involved in virulence of *S. aureus* using a silkworm model. Overall, we generated novel information advancing knowledge on platelet–*S. aureus* interaction in the unique PC storage environment, opening avenues for future studies that can contribute with knowledge in the prevention of septic transfusion reactions involving contaminated PCs.

## Figures and Tables

**Figure 1 antibiotics-14-00635-f001:**
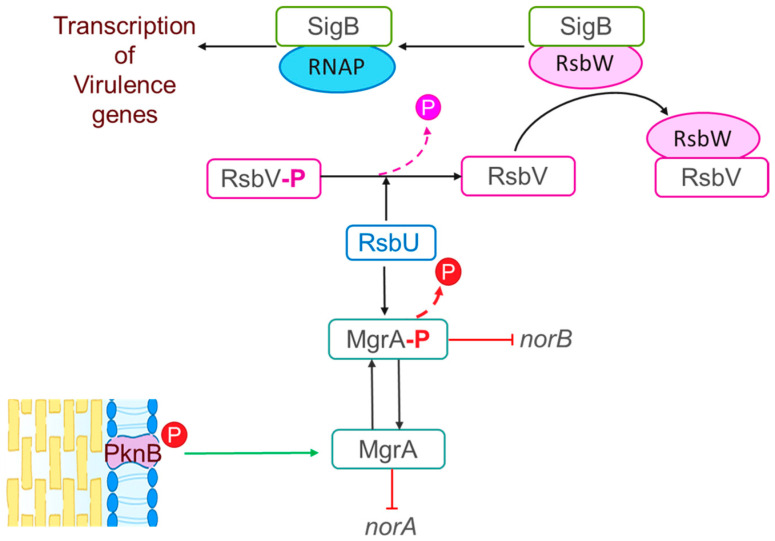
Proposed model for the regulatory cascade for the “nor” family efflux in *S. aureus*. MgrA phosphorylation is driven by the PknB kinase resulting in repression of *norA* by MgrA in its dephosphorylated state and repression of *norB* by phosphorylated MgrA (MgrA-P). MgrA can be dephosphorylated by the RsbU phosphatase, which has a major role in the alternative sigma factor SigB regulon comprising four proteins, RsbV, RsbW, RsbU, and SigB. The SigB regulon is involved in expression of virulence genes as follows: RsbU dephosphorylates RsbV-P; dephosphorylated RsbV then binds to RsbW releasing SigB to form a complex with the RNA polymerase resulting in the holoenzyme RNAP. This holoenzyme drives the transcription of virulence genes [16,20].

**Figure 2 antibiotics-14-00635-f002:**
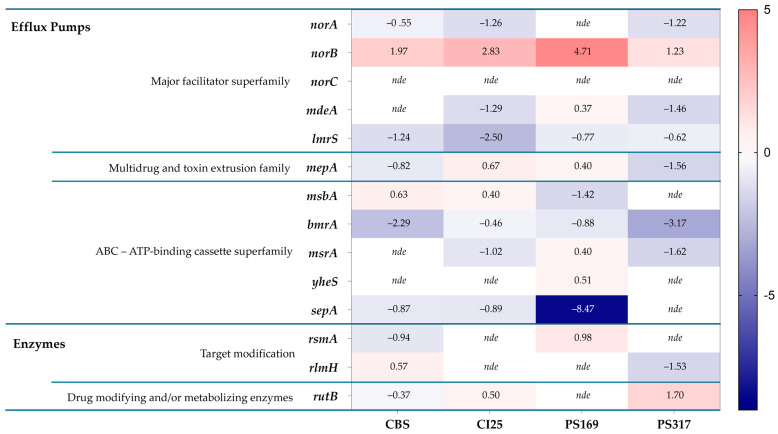
Heatmap of *S. aureus* differentially expressed genes encoding antibiotic resistance mechanisms (PCs vs. TSB) [CBS: CBS2016-05, CI25: CI/BAC/25/13/W; PS169: PS/BAC/169/17/W and PS317: PS/BAC/317/16/W]. *nde:* Not differentially expressed. Heatmap prepared with GraphPad Prism (version 9.0.0).

**Figure 3 antibiotics-14-00635-f003:**
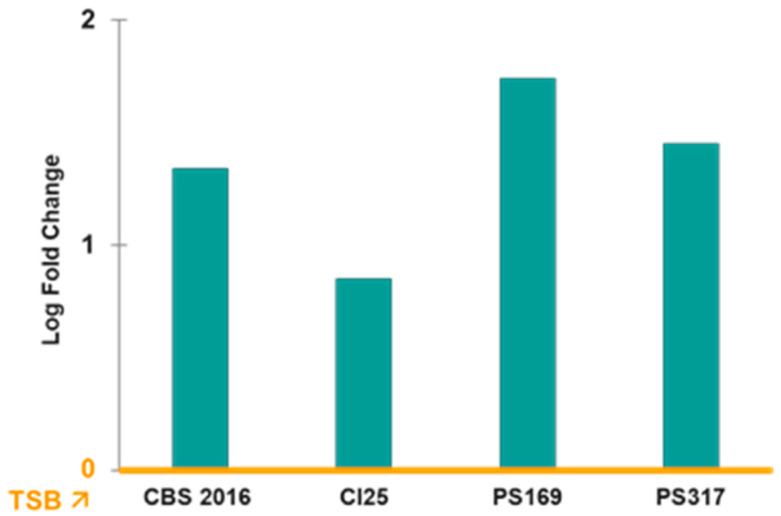
RT-qPCR of *norB* expression in PCs compared to TSB (yellow line). Relative expression of *norB* genes in transfusion-relevant *S. aureus* strains cultured in platelet concentrates (PCs) compared to trypticase soy broth (TSB). TSB is the baseline (yellow line). Gene expression levels were normalized to the housekeeping gene (16S RNA) and calculated using the ΔΔCt method. Bars represent mean fold change from three pooled replicates. Expression of *norB* in PCs versus TSB was significantly different (*p* < 0.05).

**Figure 4 antibiotics-14-00635-f004:**
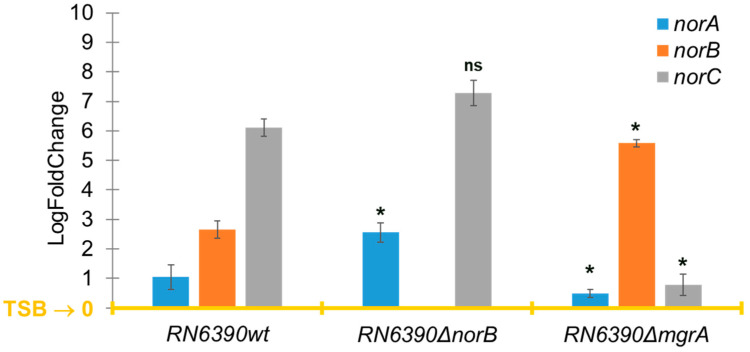
Expression of efflux pumps genes *norA*, *norB,* and *norC* measured by RT-qPCR. Relative expression of *norA*, *norB*, and *norC* in *S. aureus* RN6390 and its deletion mutants *norB* and *mgrA* cultured in platelet concentrates (PCs) compared to trypticase soy broth (TSB). TSB is the baseline (yellow line). Gene expression levels were normalized to the housekeeping gene 16SRNA and calculated using the ΔΔCt method. Bars represent mean fold-change ± standard deviation from three biological replicates. Statistical significance was determined by one-way ANOVA, * *p* < 0.05; ns: not significant.

**Figure 5 antibiotics-14-00635-f005:**
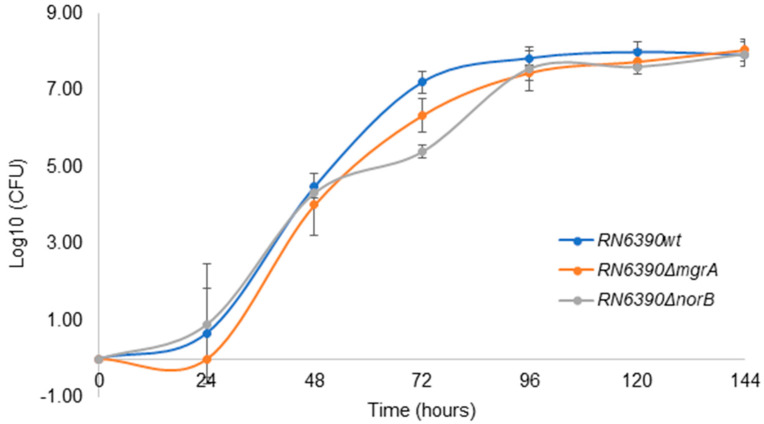
*S. aureus* RN6390, RN6390Δ*norB* and RN6390Δ*mgrA* grown in PCs. PC units inoculated with ~30 CFU/PC unit were incubated at 22 ± 2 °C, 60 rpm for 5 days (144 h). Results are presented as a mean of 3 independent trials. Error bars with standard deviation (±SD).

**Table 1 antibiotics-14-00635-t001:** List of *S. aureus* isolates investigated in this study and their origin.

Strains	Origin	Reference
Transfusion Relevant Strains (TRSs)		
CBS2016-05	PCs (involved in a septic transfusion reaction, Canada)	[26]
CI/BAC/25/13/W	PCs (involved in a near-miss case, UK)	[27]
PS/BAC/169/17/W	PCs (detected during PC screening, UK)	[28]
PS/BAC/317/16/W	PCs (detected during PC screening, UK)	[29]
Laboratory Strains		
RN6390	Wild type, descendent of NCTC8325-4 (*rsbU* mutant)	[30]
RN6390Δ*norB*	RN6390 (*norB* mutant, *rsbU* mutant)	This study
RN6390Δ*mgrA*	RN6390 (*mgrA* mutant, *rsbU* mutant)	[31]
Control Strain		
ATCC 29213	American Type Culture Collection	

**Table 2 antibiotics-14-00635-t002:** Minimal Bactericidal Concentration (MBC) of ciprofloxacin and norfloxacin in *S. aureus* grown in TSB and PCs, and statistical comparison (n > 3).

Strains	Ciprofloxacin (μg/mL)						Norfloxacin (μg/mL)		
	TSB	PCs		*p* Value *(*PCs vs. TSB*)*			TSB	PCs	*p* Value *(*PCs vs. TSB*)*
ATCC 29213	0.25–2	16–64		<0.0001			1	2–128	<0.0001
CBS2016-05 CI/BAC/25/13/W PS/BAC/169/17/W PS/BAC/317/16/W	0.5–1	1–8		0.0478			1–2	4–8	*p* > 0.05
	0.5–2	2–8		0.0303			1–4	1–4	*p* > 0.05
	1–8	8–32		0.0002			1–4	8–128	0.0571
	0.5–2	1–16		0.0571			1–2	1–16	*p* > 0.05
RN6390 RN6390Δ*norB* RN6390Δ*mgrA*	0.125–1	0.5–16		<0.0001			1	1–16	<0.0001
	0.125–0.5	0.5–4		<0.0001			0.5	1–8	<0.0001
	0.5–1	2–16		<0.0001			0.25–1	2–32	<0.0001
**Comparison between (RN6390 “wild type” and deletion mutant isolates)—Mann–Whitney U test**									
**Ciprofloxacin**			***p* value**			**Norfloxacin**			***p* value**
WT vs. Δ*norB* (PCs)			0.6501			WT vs. Δ*norB* (PCs)			0.2720
WT vs. Δ*mgrA* (PCs)			0.0002			WT vs. Δ*mgrA* (PCs)			0.0011

**Table 3 antibiotics-14-00635-t003:** LD_50_ results for wild-type and mutant *norB* and *mgrA S. aureus* strains in a silkworm animal model (n = 3, Student *t*-test *p* > 0.05).

*S. aureus* Isolates	LD_50_ (Per Larvae)
RN6390 “wild type” (WT)	~1.02 × 10^4^ CFU (±1.01 × 10^4^)
RN6390Δ*norB*	~3.29 × 10^6^ CFU (±2.04 × 10^6^)
RN6390Δ*mgrA*	~2.85 × 10^5^ CFU (±1.90 × 10^5^)
**Comparison between strains**	** *T* ** **-test with Welch’s correction (*p* value)**
WT vs. Δ*norB*	0.1518
WT vs. Δ*mgrA*	0.1773

## Data Availability

The data of all results in this study are included in the manuscript. Raw data will be provided by the corresponding author upon reasonable request.

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
