# Peer review of "Enhanced Quinolone Resistance and Differential Expression of Efflux Pump nor Genes in Staphylococcus aureus Grown in Platelet Concentrates"

_antibiotics, 2025, doi:10.3390/antibiotics14070635_

Round 1
Reviewer 1 Report
Comments and Suggestions for Authors
This study investigated how the platelet concentrate (PC) storage environment affects Staphylococcus aureus, revealing that PC conditions induce strain-dependent upregulation of nor efflux pump genes and increase resistance to quinolone antibiotics. Notably, the efflux pump gene norB was also linked to enhanced virulence in a silkworm model, highlighting potential risks for septic transfusion reactions from S. aureus-contaminated PCs. The study is interesting and well executed; my concerns are minor and can be addressed with the following revisions:
1. Please correct all instances of “16s RNA” to the correct term “16S rRNA” throughout the manuscript.
2. Line 332: Provide the full species name for ATCC 29213 (e.g., Staphylococcus aureus ATCC 29213) for clarity.
3. In Table 2, the legend should define the abbreviation “nde” (e.g., “not differentially expressed”) for the reader’s understanding.
4. The Discussion should acknowledge the limitation of the silkworm model: while suitable for preliminary virulence screening, it does not replicate the complexities of human transfusion physiology or immune responses. The authors should note that these findings require confirmation in human cell lines or mammalian models to validate clinical relevance.
Author Response
REVIEWER 1
This study investigated how the platelet concentrate (PC) storage environment affects Staphylococcus aureus, revealing that PC conditions induce strain-dependent upregulation of nor efflux pump genes and increase resistance to quinolone antibiotics. Notably, the efflux pump gene norB was also linked to enhanced virulence in a silkworm model, highlighting potential risks for septic transfusion reactions from S. aureus-contaminated PCs. The study is interesting and well executed; my concerns are minor and can be addressed with the following revisions:
- Please correct all instances of “16s RNA” to the correct term “16S rRNA” throughout the manuscript.
ANSWER: Changes have been made as requested.
- Line 332: Provide the full species name for ATCC 29213 (e.g., Staphylococcus aureus ATCC 29213) for clarity.
ANSWER: Changes have been made as requested.
- In Table 2, the legend should define the abbreviation “nde” (e.g., “not differentially expressed”) for the reader’s understanding.
ANSWER: Table 2 has been converted in a Heatmap (new Figure 2 of the revised manuscript).
- The Discussion should acknowledge the limitation of the silkworm model: while suitable for preliminary virulence screening, it does not replicate the complexities of human transfusion physiology or immune responses. The authors should note that these findings require confirmation in human cell lines or mammalian models to validate clinical relevance.
ANSWER: We appreciate the Reviewer’s comment and accordingly, a new section entitled “Recommendations for further investigation” has been added to the revised manuscript highlighting the limitations of the silkworm model and need to validate the data with cell lines or mammal animal models.
Reviewer 2 Report
Comments and Suggestions for Authors
Staphylococcus aureus grown in platelet concentrates displays enhanced quinolone resistance with differential expression of efflux pump nor genes in a strain-dependent manner
Based on the expression profile, the current study demonstrates that Staphylococcus aureus in platelet concentrates develops resistance to quinolones primarily through the overexpression of efflux pumps. The study primarily interested and manuscript is well organized, however some contents and information are needed to improve in scientific level.
Following requests are necessary to be addressed to improve quality of the study.
Title:
- In my opinion the title need to shorter than this.
- In my opinion, the experiment you performed is not sufficient to conclusively demonstrate that the differential expression of efflux pump nor genes occurs in a strain-dependent manner.
Abstract:
- Line 14 , did you study the effect impact of PC storage on virulence factors ? .
- Line 15 , It will be clearer to explain why you chose quinolone.
- Line 17 , what you mean by “ These studies revealed differentially expressed genes involved in antibiotic resistance” which studies. Need to double check .
- Line 32 , You need to include the limitations of the study and suggest directions for future research.
Introduction:
- In this section, you need to include a paragraph describing the possible mechanisms by which Staphylococcus aureus resists quinolones.
- Line 100 , it is necessary to elucidate the specific mechanism by which the PC environment up-regulates virulence factors or resistance genes, including the norA gene.
Results:
- Line 112: The antibiotic resistance profile of these strains should be thoroughly evaluated prior to their use in the study, with particular emphasis on their ability to resist quinolones.
- Line 113 :Could you please explain the source of these strains?
- Line 121 : Is the change in norb from 1.23 to 4.71 statistically significant?
- Line 133 , NorA and NorC were upregulated in the Rna630 strain, while only NorB is upregulated in its derivative mutant (MgrA). If MgrA specifically regulates NorB, why are NorA and NorC affected without being upregulated?
- Line 185 , did you test your hypothesis ?
- Line 168 , How norB enhanced virulence of S.aureus ?
- Line 163, Which virulence factor in the selected strain is potentially enhanced by NorB? Please specify and provide an explanation.
- Line 188 ,Figure 2: There is no indication of statistical significance.
- Line 208 , Table 2 displays the results for MATE and ABC transporter, but these are not mentioned in the Results section.
Discussion
-This section should clearly articulate the impact of the study and specify how it contributes to and advances previous research in the field.
Methods:
- Did this study perform whole genome sequencing of the selected strain? If not, how were the genomes of the different strains compared?
- Since most of the results rely on qPCR, why was only one housekeeping gene used? Wouldn't using 3 to 4 housekeeping genes enhance precision?
- The growth rate of bacteria in TSB differs from that in PC, which may potentially affect the qPCR results and total RNA yield, potentially leading to an unfair comparison.( explain )
- What is the bacterial concentration used to initiate the experiment?
Finally, all above, the manuscript is need to minor revision
Author Response
REVIEWER 2
Staphylococcus aureus grown in platelet concentrates displays enhanced quinolone resistance with differential expression of efflux pump nor genes in a strain-dependent manner
Based on the expression profile, the current study demonstrates that Staphylococcus aureus in platelet concentrates develops resistance to quinolones primarily through the overexpression of efflux pumps. The study primarily interested and manuscript is well organized, however some contents and information are needed to improve in scientific level.
Following requests are necessary to be addressed to improve quality of the study.
Title:
- In my opinion the title need to shorter than this.
ANSWER: The title of the revised manuscript has been shortened as recommended.
- In my opinion, the experiment you performed is not sufficient to conclusively demonstrate that the differential expression of efflux pump nor genes occurs in a strain-dependent manner.
ANSWER: Transcriptome data and RT-qPCR experiments demonstrated that nor genes are differentially expressed in the strains tested in our study when grown in platelet concentrates vs TSB. We agree that we cannot generalized our findings to all S. aureus strains and therefore “strain-dependent” has been removed from the title and conclusion of the revised manuscript.
Abstract:
- Line 14 , did you study the effect impact of PC storage on virulence factors ? .
ANSWER: Thank you for this observation. No, in this study we do not have data of the effect of PC on virulence. The error has been corrected in the abstract of the revised version of the manuscript.
- Line 15 , It will be clearer to explain why you chose quinolone.
ANSWER: Thank you for this observation. Rationale for testing quinolones, which are commonly used to treat S. aureus infections in Canada, has been added to the abstract of the revised manuscript.
- Line 17 , what you mean by “ These studies revealed differentially expressed genes involved in antibiotic resistance” which studies. Need to double check .
ANSWER: Wording has been changed from “These studies” to “Results of these analyses” to clarify that we are referring to the transcriptome analyses mentioned in the previous sentence.
- Line 32 , You need to include the limitations of the study and suggest directions for future research.
ANSWER: Thank you for this observation. Limitations of the silkworm model to study virulence of S. aureus has been addressed in Conclusions of the abstract of the revised manuscript.
Introduction:
- In this section, you need to include a paragraph describing the possible mechanisms by which Staphylococcus aureus resists quinolones.
ANSWER: Thank you for this observation. Information explaining the mechanisms of quinolone resistance, target alteration and efflux pumps, has been added to the 4th paragraph of the introduction of the revised manuscript.
- Line 100 , it is necessary to elucidate the specific mechanism by which the PC environment up-regulates virulence factors or resistance genes, including the norA gene.
ANSWER: Thank you for this observation. The specific molecular mechanisms by which the PC storage environment modulates bacterial gene expression are unknown and this information has been added in the second last paragraph of the introduction of the revised manuscript. However, we do not know that S. aureus induces changes in platelet activation and expression of immune factors which we have previously published and referred to in the present manuscript (e.g., Chi and Ramirez-Arcos, Ann Blood 2025, 10, 5–5, doi:10.21037/aob-24-31). These platelet changes in return affect bacterial gene expression, but we do not know how this regulation is done at the molecular level.
Results:
- Line 112: The antibiotic resistance profile of these strains should be thoroughly evaluated prior to their use in the study, with particular emphasis on their ability to resist quinolones.
ANSWER: Thank you for this observation. The goal of the study was not to characterize the antibiotic resistance of the strains but to compare resistance to two specific quinolones between bacterial grown in media versus platelets. Therefore, we only tested two quinolones, and the MIC data have now been included in the revised manuscript as supplementary table S1 in the revised manuscript. As the goal was to compare antibiotic resistance between media and platelet concentrates, Minimal Bactericidal Concentration was more appropriate due to the viscosity of platelet products and therefore that was the focus of the experiments.
- Line 113 :Could you please explain the source of these strains?
ANSWER: The origin of the strains has been added to the text in the first paragraph of Results and Table 1 of the revised manuscript.
- Line 121 : Is the change in norb from 1.23 to 4.71 statistically significant?
ANSWER: As stated in Materials and Methods of the revised manuscript, section 4.4, a log2 fold difference of ≥1.0 in the transcriptome data was considered a significant change in DGE. This information has been added in the second paragraph of the Results section 2.1 of the revised manuscript.
- Line 133 , NorA and NorC were upregulated in the Rna630 strain, while only NorB is upregulated in its derivative mutant (MgrA). If MgrA specifically regulates NorB, why are NorA and NorC affected without being upregulated?
ANSWER: Thank you for this comment. The regulation of nor genes is a complex process with different players, which can also be differentially regulated in PCs in different strains such as the mgrA mutant. Lack of expression of norA and norC in the mgrA mutant could be due to the action of other regulators such as NorG and ArlR-ArlS. Information explaining this hypothesis with supporting references has been added in the second paragraph of section 2.1 of the Results in the revised manuscript.
- Line 185 , did you test your hypothesis ?
ANSWER: In the Results section 2.3 we hypothesized that NorB could be involved in increased growth in PCs and enhanced virulence of S. aureus. While our results showed that NorB does not impact growth of S. aureus in PCs, the efflux pump is involved in virulence as shown with experiments using silkworm larvae. Paragraphs 2 and 3 of Section 2.3 have been edited to address this comment from the Reviewer in the revised manuscript
- Line 168 , How norB enhanced virulence of S.aureus ?
ANSWER: We propose that the efflux pump exports immune factors such as antimicrobial peptides that are released by silkworms as response to the infection with S. aureus. This explanation has been added to paragraph 3 of Section 2.3 of the revised manuscript with a supporting reference.
- Line 163, Which virulence factor in the selected strain is potentially enhanced by NorB? Please specify and provide an explanation.
ANSWER: NorB does not induce expression of virulence factors; this efflux pump likely enhances virulence by exporting antimicrobial peptides from the staphylococcal cells.
- Line 188 ,Figure 2: There is no indication of statistical significance.
ANSWER: Fold difference of norB expression in platelet concentrates compared to TSB was significant different. P<0.05 was indicated in the legend of the Y axis in figure 2 of the original version of the manuscript. In the revised manuscript, this is figure 3 and the p value has been added to the figure legend.
- Line 208 , Table 2 displays the results for MATE and ABC transporter, but these are not mentioned in the Results section.
ANSWER: Genes of these families were differentially expressed as stated at the end of the first paragraph of section 2.1 of the results.
Discussion
-This section should clearly articulate the impact of the study and specify how it contributes to and advances previous research in the field.
ANSWER: Thank you for this comment. Our study advanced knowledge providing evidence that S. aureus grown in PCs has increased antibiotic resistance. Furthermore, we provided proof of the involvement of NorB in S. aureus virulence. These statements have been added to the first paragraph of the discussion of the revised manuscript.
Methods:
- Did this study perform whole genome sequencing of the selected strain? If not, how were the genomes of the different strains compared?
ANSWER: We have previously performed and published whole genome sequencing of the transfusion relevant strains (please refer to references 28-31 of the manuscript). Strain RN3690 was previously sequenced and published (Ref 37). These references are cited in the manuscript and in Table 1. Also, we have added a new section (7.2) in Materials and Methods of the revised manuscript describing the methods for the comparative genomic analyses.
- Since most of the results rely on qPCR, why was only one housekeeping gene used? Wouldn't using 3 to 4 housekeeping genes enhance precision?
ANSWER: The main gene expression results and focus of our study with transfusion relevant strains was obtained with transcriptome analyses. RT-qPCR was done either to validate transcriptome results or to explain Minimal Bactericidal Concentration data of laboratory strains. For RT-qPCR, we followed the same approach as the one used in previous work published by our team with the gyrA gene as the only housekeeping gene: Yousuf, et al. PLoS ONE 2024, 19, e0307920, doi:10.1371/journal.pone.0307920
- The growth rate of bacteria in TSB differs from that in PC, which may potentially affect the qPCR results and total RNA yield, potentially leading to an unfair comparison.( explain )
ANSWER: We ensure that RNA was extracted at the same growth phase independently of the growth rate. This is a standard procedure in our laboratory as it is recognized that bacteria have different growth rates in media compared to PCs. For the experiments of the present study, TSB and PC cultures were both grown until stationary phase prior to RNA extraction. The RNA libraries were then normalized and diluted as needed for sequencing, following previous published protocols (Yousuf, et al. PLoS ONE 2024, 19, e0307920, doi:10.1371/journal.pone.0307920). Further details regarding RNA extraction and normalization have been provided in Materials and Methods section 4.4 in the revised manuscript.
- What is the bacterial concentration used to initiate the experiment?
ANSWER: The initial inoculum was 4 x 106 CFU/mL and this information has been added to section 4.4 of Materials and Methods of the revised manuscript.
Finally, all above, the manuscript is need to minor revision
Reviewer 3 Report
Comments and Suggestions for Authors
This is a beautiful piece of work, and I enjoyed reading through it, especially the findings. However, some areas urgently need improvement to make the work more appealing to readers and better interpreted.
1) In your abstract (line 14), can authors explicitly state the objective of the study? Authors implied but did not explicitly state the objective of the study. In line 25 “norA and norC may be involved in” is vague. What is the determinant of this inference?
In line 31 the phrase “inform clinicians”. To me, sounds common. Why don’t Authors clearly say example; "our finding provides mechanistic insight that may/will inform antimicrobial selection in cases of S. aureus contaminated platelate transfusion ……….…………….. or something more strong. The Authors sounded like they are not sure or confident of their findings.
2) In your introduction, Authors relied heavily on previous CBS studies. If you see Lines 49-52 and 94 to 100. There is a repetition on previous CBS. I think the Authors should arrange the introduction section, reduce repetition of previous CBS studies, give the background of the study, justify why the study was carried out, its significance, central hypothesis and gaps and offcourse what we stand to gain in the introduction section.
3) In the result section, line 123: The phrase “not differentially expressed” for norA/norC is unclear; can you use precise values or “log2 FC ~0”. .
-See line 135-136, the strain depe ndent adaptative is not explained in details. I am struggling to understand.
-Also in the result section, values are not reported when statistical inferences are concluded or made. eg. see 147-150.
-There are some parts of the result interpretation sections where no statistical significance was found for some comparisons but these are still interpreted (see 165-167). Please Authors should be careful while interpreting results that are not statistically significant. Instead of doing that, we can adopt Bayesian theory where we can incorporate prior.
-I will also suggest that Authors should discuss why RN6390 behaves differently- mention rsbU mutation earlier and clearly.
-I will suggest that table 2, be presented as a heatmap for better visualization.
-I couldn’t find any MIC table. This is standard protocol in assessing bactericidal vs bacteriostatic.
-Authors should include primers efficiency curve in the result section , if possible, because in the real sense, if there are not curve of efficiency, ΔΔCt values may be unreliable.
4) In your discussion section, see line 240- 246, I am not very comfortable how Authors explained regulatory interaction, it sounds to speculative. Can Authors revisit please.
-See Line 259, the rsbU negative background, in my opinion, needs to be linked better to phenotypic differences.
5) In your methodology section, there are double 4.6. Please confirm.
-I am not sure I saw anything like Primer sequences for RT-qPCR in the body of the work or a reference of S2 in the supplimentary file.
-If you ask me, I will say n is not well defined for RNA-seq experiments. Because the number of biological replicates is missing. In real life, RNA-seq without sufficient replicates will lack statistical power to identify true DEGS. (this limitation can be incorporated into the limitations section.
6) In your conclusion, I think there are overstatement of clinical translation. Considering the fact that there in-vivo mammalian was not studied. A statement of further studies using mamamalian models in the conclusion section, will be good.
7) Generally, I think there are a lot of limitaitions in this study, that warrants a section called “Limitations of the study” where all the issues like lack of mammalian host data….. etc. can go in.
Author Response
REVIEWER 3
This is a beautiful piece of work, and I enjoyed reading through it, especially the findings. However, some areas urgently need improvement to make the work more appealing to readers and better interpreted.
1) In your abstract (line 14), can authors explicitly state the objective of the study? Authors implied but did not explicitly state the objective of the study. In line 25 “norA and norC may be involved in” is vague. What is the determinant of this inference?
ANSWER: Thank you for this comment. The aim of the study has been explicitly stated in the abstract of the revised manuscript in the Background/Objective section.
In line 31 the phrase “inform clinicians”. To me, sounds common. Why don’t Authors clearly say example; "our finding provides mechanistic insight that may/will inform antimicrobial selection in cases of S. aureus contaminated platelate transfusion ……….…………….. or something more strong. The Authors sounded like they are not sure or confident of their findings.
ANSWER: We appreciate this suggestion and the conclusion of the abstract of the revised manuscript has been edited accordingly.
2) In your introduction, Authors relied heavily on previous CBS studies. If you see Lines 49-52 and 94 to 100. There is a repetition on previous CBS. I think the Authors should arrange the introduction section, reduce repetition of previous CBS studies, give the background of the study, justify why the study was carried out, its significance, central hypothesis and gaps and offcourse what we stand to gain in the introduction section.
ANSWER: Thank you for this comment. The introduction of the revised manuscript has been edited accordingly.
3) In the result section, line 123: The phrase “not differentially expressed” for norA/norC is unclear; can you use precise values or “log2 FC ~0”. .
ANSWER: Clarification of “not differentially expressed” has been added in brackets: “(i.e., log2 fold change = 0)” in the revised manuscript.
-See line 135-136, the strain depe ndent adaptative is not explained in details. I am struggling to understand.
ANSWER: In response to this comment, the last sentence of section 2.1 has been edited to read” “The differences observed in nor gene expression between TRS and RN6390 strains indicate that gene regulation of S. aureus grown in PCs depends on the strain’s genetic background.”
-Also in the result section, values are not reported when statistical inferences are concluded or made. eg. see 147-150.
ANSWER: p values for all comparisons are disclosed in Tables 2 and 3 of the revised manuscript. Furthermore, information about these values has been added where appropriate in the text of the Results sections 2.2 and 2.3 of the revised manuscript.
-There are some parts of the result interpretation sections where no statistical significance was found for some comparisons but these are still interpreted (see 165-167). Please Authors should be careful while interpreting results that are not statistically significant. Instead of doing that, we can adopt Bayesian theory where we can incorporate prior.
ANSWER: For some of the results it is important to analyze the data beyond statistical significance. We acknowledge that the LD50 for the silkworm assays was not statistically different between wild-type and the norB mutant strain. However, 2log of bacteria is biologically meaningful, especially when each set of experiments was done with 10 larvae with three independent replicates, which consistently showed larvae death at specific bacterial loads. A better explanation of the interpretation of our results has been added to section 2.3 of the Results of the revised manuscript.
-I will also suggest that Authors should discuss why RN6390 behaves differently- mention rsbU mutation earlier and clearly.
ANSWER: Alteration of the phosphorylation status of MgrA and consequent differential regulation of nor genes as result of the rsbU mutation has been added to the last sentence of the first paragraph of section 2.2 of the results in the revised manuscript.
-I will suggest that table 2, be presented as a heatmap for better visualization.
ANSWER: We appreciate this suggestion and therefore Table 2 ahs been converted to a heatmap.
-I couldn’t find any MIC table. This is standard protocol in assessing bactericidal vs bacteriostatic.
ANSWER: The goal of the study was not to compare bactericidal vs bacteriostatic effect of the quinolones but to compare resistance to these antibiotics when S. aureus was grown in PCs versus TSB. For this purpose, MBC is the assay that was chosen due to the turbidity of PCs which would not allow to compare MIC data. However, we did perform MIC along with MBC in media and that data have now been added to the revised manuscript. MIC is presented in a new supplementary table (Table S1).
-Authors should include primers efficiency curve in the result section , if possible, because in the real sense, if there are not curve of efficiency, ΔΔCt values may be unreliable.
ANSWER: Primer efficiency was tested and is now included in section 4.5 and Supplementary figures S2 and S3 of the revised manuscript.
4) In your discussion section, see line 240- 246, I am not very comfortable how Authors explained regulatory interaction, it sounds to speculative. Can Authors revisit please.
ANSWER: Paragraph 4 of the discussion of the revised manuscript has been edited to address the Reviewer’s concern. Our data showing differences in nor gene expression between strains that carry rsbU and rsbU mutants strongly suggest that the phosphorylation status of the global regulator MgrA is affected. However, further experiments are needed to confirm this hypothesis, which are beyond the scope of this study but are proposed in the new section entitled “Recommendations for further investigation” (added to the revised manuscript in response to this Reviewer’s suggestion).
-See Line 259, the rsbU negative background, in my opinion, needs to be linked better to phenotypic differences.
ANSWER: Additional discussion of virulence driven by norB in rsbU- vs rsbU+ strains has been added to the second last paragraph of the Discussion of the revised manuscript.
5) In your methodology section, there are double 4.6. Please confirm.
ANSWER: Thank you for this observation. This error has been corrected as section 4.5 was mislabeled as section 4.6.
-I am not sure I saw anything like Primer sequences for RT-qPCR in the body of the work or a reference of S2 in the supplimentary file.
ANSWER: Thank you for this observation. This error has been corrected, primer sequences for RT-qPCR are listed in supplementary Table S2 of the revised manuscript.
-If you ask me, I will say n is not well defined for RNA-seq experiments. Because the number of biological replicates is missing. In real life, RNA-seq without sufficient replicates will lack statistical power to identify true DEGS. (this limitation can be incorporated into the limitations section.
ANSWER: Information on the number of replicates (3 independent repetitions) has now been added to section 4.4 of Materials and Methods of the revised manuscript.
6) In your conclusion, I think there are overstatement of clinical translation. Considering the fact that there in-vivo mammalian was not studied. A statement of further studies using mamamalian models in the conclusion section, will be good.
ANSWER: This statement has been added in the “Recommendations for further investigation” new section of the revised manuscript.
7) Generally, I think there are a lot of limitaitions in this study, that warrants a section called “Limitations of the study” where all the issues like lack of mammalian host data….. etc. can go in.
ANSWER: We agree, and a new “Recommendations for further investigation” section has been added in the revised manuscript.
Round 2
Reviewer 3 Report
Comments and Suggestions for Authors
The authors have thoroughly addressed all the comments and suggestions raised during the review process.